# Pulmonary Histoplasmosis Mimicking Metastatic Lung Cancer: A Case Report

**DOI:** 10.3390/diagnostics11020328

**Published:** 2021-02-17

**Authors:** Gion Ruegg, Stefan Zimmerli, Maria Trachsel, Sabina Berezowska, Swantje Engelbrecht, Yonas Martin, Martin Perrig

**Affiliations:** 1Department of General Internal Medicine, Inselspital, Bern University Hospital, University of Bern, 3010 Bern, Switzerland; maria.trachsel@insel.ch (M.T.); yonas.martin@insel.ch (Y.M.); martin.perrig@insel.ch (M.P.); 2Department of Intensive Care Medicine, University Hospital, University of Bern, 3010 Bern, Switzerland; 3Department of Infectious Diseases, Inselspital, Bern University Hospital, University of Bern, 3010 Bern, Switzerland; stefan.zimmerli@insel.ch; 4Institute of Pathology, University of Bern, 3008 Bern, Switzerland; sabina.berezowska@chuv.ch; 5Department of Laboratory Medicine and Pathology, Institute of Pathology, Lausanne University Hospital, University of Lausanne, 1011 Lausanne, Switzerland; 6Department of Nuclear Medicine, Inselspital, Bern University Hospital, University of Bern, 3010 Bern, Switzerland; swantje.engelbrecht@insel.ch

**Keywords:** histoplasma capsulatum, PET-CT, flip-flop fungus sign, pulmonary lesion, lymphadenopathy, necrotic granulomatosis

## Abstract

Histoplasmosis is a well-known endemic fungal infection but experience in non-endemic regions is often limited, which may lead to delayed diagnosis and extensive testing. The diagnosis can be especially challenging, typically when the disease first presents with pulmonary nodules accompanied by hilar and mediastinal lymphadenopathy, suggesting a much more common malignant disease. In this situation, a greater FDG uptake in draining lymph nodes in comparison with the associated lung nodule seen in [^18^F]FDG-PET/CT, the so-called “flip-flop fungus” sign, can help to orientate further diagnostic measures. We report a case of a 56-year-old woman living in Switzerland, a non-endemic region, whose diagnosis of imported histoplasmosis was delayed since the findings had been initially misinterpreted as pulmonary malignancy. Further, histological workup was inconclusive due to lack of specific fungal staining, leading to ineffective treatment and non-resolving disease. This paper intends to highlight the pitfalls in diagnosing *Histoplasma capsulatum* and presents images of particularities of fungal infections in [^18^F]FDG-PET/CT, which in our case showed a “flip-flop fungus” sign.

## 1. Introduction

We report a case of pulmonary histoplasmosis in a Swiss woman, who had travelled to Costa Rica, a region where *Histoplasma capsulatum* is endemic, and in whom the initial computed tomography scan (CT) showed a pulmonary lesion with mediastinal and hilar lymphadenopathy compatible with pulmonary malignancy [1]. The following 2-deoxy-2-[^18^F]fluoro-D-glucose positron emission tomography/computed tomography ([^18^F]FDG-PET/CT) showed a higher fluoro-D-glucose (FDG) uptake in the draining lymph nodes than in the associated lung nodule, a so-called “flip-flop fungus” sign, first described by Nagelschneider [2]. A following supraclavicular lymph node resection eventually led to the diagnosis of histoplasmosis.

Histoplasmosis is a fungal infection caused by *H. capsulatum complex*, which is mostly asymptomatic but can also cause severe and disseminated disease. Transmission usually occurs through inhaled conidia or mycelial fragments from contaminated soil, where *H. capsulatum* grows best when it is enriched with bird or bat droppings [3]. Diagnostic modalities include antigen testing, serology (immunodiffusion and complement fixation), polymerase chain reaction assays, cultures and histopathologic studies [4,5,6,7]. Although histoplasmosis is a common endemic disease throughout the world, its incidence in Western Europe is very low. Recent epidemiological data from Switzerland are lacking, but a study conducted in 1998–99 reported only four cases over these two years [8].

Since experience with *H. capsulatum* infections is often limited in non-endemic regions, the disease is likely to be misdiagnosed as more common conditions, like pulmonary malignancy, autoimmune granulomatous diseases, or mycobacterial infections. Our report aims to share knowledge of clinical, radiological and histopathological findings in this fungal infection to help prevent misdiagnosis.

## 2. Case

A 56-year-old woman consulted her general practitioner (day 0) for chest pain and shortness of breath without fever or cough, which she had developed about three weeks after her return (day -40) from a three-week holiday in Costa Rica. During her holidays, she visited dusty rural areas where birds and bats were sighted, but without any close encounters with these animals. The patient was born in Switzerland, has never traveled outside of Europe before and was an office worker without expositional hazards. Medical history was significant for undifferentiated connective tissue syndrome, treated with hydroxychloroquine 200 mg/d for the past eight years, mild haemophilia type A and fatty liver disease. Clinical examination, in particular, cardiopulmonary status, was unremarkable. Laboratory tests revealed moderately elevated C-reactive protein (43 mg/L), elevated liver function tests (alanine aminotransferase (ALT) 3 times, aspartate transaminase (AST) 2 times the upper limit) and increased erythrocyte sedimentation rate (40 mm/h). The full blood count was normal. Initial diagnostic workup ruled out an acute coronary syndrome, whereas a CT scan of the chest (day +7) excluded pulmonary embolism but revealed a spiculated lesion of 22 mm in diameter in the right lower lobe of the lung with right hilar and mediastinal lymphadenopathy, highly suspicious for malignancy. CT-guided core needle biopsy (day +9, complicated by pneumothorax and the need for a chest tube) obtained only small samples of the pulmonary lesion showing organizing pneumonia on histopathological work-up. No malignant infiltrates, no bacteria and no fungi could be identified, even upon applying special stains for infectious organisms (Grocott’s methenamine silver stain).

To further evaluate the suspected malignant disease, a [^18^F]FDG-PET/CT was performed (day +14). It detected multiple hypermetabolic lymph nodes of up to 33 mm in diameter, located right interlobar, at the right hilus, the mediastinum and right supraclavicular. The previously known lesion of the right lung showed only moderate metabolic activity. In addition, a moderately active precaval lymph node was detected (Figure 1).

Histopathological work-up after excision of two supraclavicular lymph nodes (day +18) revealed granulomatous lymphadenitis with necrosis. Although no pathogens could be visualized (on Gram, Ziehl-Neelsen, Periodic Acid Shift (PAS) and Giemsa stain), the image was considered to be most consistent with an infectious disease.

Further workup with bronchoalveolar-lavage (BAL, day +30) revealed alveolar lymphocytosis with a normal CD4/CD8-T-cell ratio without any growth of bacteria, mycobacteria (incl. *M. tuberculosis* after eight weeks) or fungi (after two days) in the cultures or any histological evidence of malignant cells. Serologies were negative for Human Immunodeficiency Virus, *Brucella spp*., *Coxiella burnetii*, *Franciscella tularensis*, *Bartonella henselae* and autoimmune diseases (anti-nuclear antibodies, anti-neutrophil cytoplasmic antibodies).

After approximately one month of outpatient diagnostic procedures, having yielded no evidence of malignancy or identifiable infection, autoimmune granulomatous disease was assumed. Subsequently, an immunosuppressive treatment with high dose corticosteroids (prednisolone 1 mg/kg body weight daily) was initiated (day +32). After initial improvement of the clinical symptoms, the patient deteriorated again over the next month. This led to the patient being referred to the general internal medicine outpatient clinic at Bern University Hospital (day +63).

Histopathological conciliary reevaluation with additional application of Grocott’s methenamine silver stain on the resected lymph nodes detected micrometric (~5µm) ovoid elements typical of *H. capsulatum* in the necrotic areas (Figure 2c), which were not visible in the PAS stain. Although amplification of DNA for sequencing of the internal transcribed spacer region from paraformaldehyde-fixated biopsy tissue, serology and urinary antigen was negative for *H. capsulatum* (day +63), pulmonary histoplasmosis was diagnosed based on the pathognomonic histomorphological results.

Consecutive laboratory testing revealed rising liver enzymes (ALT up to 14 times, AST 5 times, alkaline phosphatase 2 times, gamma-glutamyl transferase 15 times upper limit), with negative serology for viral hepatitis A, B, C and E (day +65). Sonographic imaging of the liver (day +70) was consistent with previously known liver steatosis. Further evaluation with liver biopsy was refused by the patient.

Since first-line antifungal agents against *H. capsulatum* are well known for their potential hepatotoxicty, and spontaneous recovery from histoplasmosis in this otherwise immunocompetent patient was to be expected, no antifungal treatment was initiated, and the corticosteroids were tapered down [9]. Over the next month, the patient’s symptoms and liver function tests improved consecutively, and so she was discharged into further care by her general practitioner (day +93) and showed no signs of a recurrent or persisting infection after one year.

## 3. Discussion

Although histoplasmosis is well-known in endemic regions, it remains rare in Western Europe and, therefore, often poses a diagnostic challenge. As presented in our case, the radiological appearance of histoplasmosis bears resemblance to metastatic lung cancer or autoimmune conditions, which may lead to extensive testing and/or wrong treatment strategy.

Differential diagnoses of non-malignant pulmonary lesions with lymphadenopathy include fungal infections, like histoplasmosis or coccidioidomycosis, and bacterial infections, such as tuberculosis, non-tuberculous mycobacteriosis, tularemia, melioidosis or brucellosis. In addition, non-infectious inflammatory diseases, like sarcoidosis, granulomatosis with polyangiitis (Wegener’s granulomatosis), eosinophilic granulomatosis with polyangiitis (Churg-Strauss syndrome) and pulmonary lymphomatoid granulomatosis, are to be considered [10]. A detailed travel history with epidemiological knowledge of the destination can substantially contribute to adapt the spectrum of differential diagnoses.

There are various diagnostic methods for histoplasmosis and the context of clinical presentations needs to be considered. Antigen or antibody testing in urine and blood provides the highest sensitivity during the acute phase of the disease (up to 80%). Antigen testing generally has higher sensitivity in diffuse disease due to higher fungal burden than in localized disease where sensitivity is only ~40%. In the latter cases, serological tests for antibodies using both immunodiffusion and complement fixation are recommended, with a reported sensitivity of ~90%. It is important to note that antibodies usually appear in the second month after transmission and may be falsely negative in immunosuppressed individuals. Culture and antigen tests from BAL fluid may help in detecting *H. capsulatum* in chronic disease [11].

The diagnostic contribution of [^18^F]FDG-PET/CT has been demonstrated in invasive fungal infections, especially in detecting occult foci. [^18^F]FDG-PET/CT is capable of detecting increased metabolic activity, indicative of inflammatory cell activity, prior to the onset of anatomical abnormalities detectable by conventional radiological means. However, discriminating infection from malignancy and other inflammatory diseases remains difficult [12,13,14]. In our patient, [^18^F]FDG-PET/CT revealed intense metabolic activity in intrathoracic and supraclavicular lymph nodes. In contrast, the pulmonary lesion in the right lobe displayed only moderate metabolic activity. This finding of equal or greater FDG uptake in draining lymph node(s) in comparison with the associated lung nodule was described by Nagelschneider as “flip-flop fungus sign” (Table 1) with a sensitivity of 60% and specificity of 85% for benign fungal disease, predominately histoplasmosis in a study population from an endemic area [2]. Another approach by Kadaria suggested that dual time point [^18^F]FDG-PET/CT can be a helpful diagnostic tool for discriminating between benign and malignant lesions in areas endemic for *H. capsulatum* [15]. The latter radiologic technique was not used in our case. Although useful, [^18^F]FDG-PET/CT is not recommended in routine diagnostics of fungal infections, mainly due to high costs and generally low availability.

In our opinion, some aspects of the present case need to be addressed explicitly. First, the precaval lymph node with moderately elevated FDG uptake and the concomitant hepatitis remain unexplained but, in retrospect, could be attributed to hepatic involvement in histoplasmosis, which may occur in immunocompetent patients on rare occasions [16]. Second, the negative serological and molecular markers in our patient may be explained by the delay in diagnosis to a time when the infection was already resolving, the poor sensitivity of these tests in non-disseminated disease and the intermittent immunosuppressive therapy with corticosteroids. Third, the initial histopathologic evaluation of the resected lymph nodes with Ziehl-Neelsen, PAS and Giemsa stains revealed necrotic granuloma but failed to identify a fungal pathogen. Although certain phases of *H. capsulatum* can be visualized by these stains, silver stains, such as Grocott’s methenamine silver stain, are considered the most reliable for the detection of *H. capsulatum* [7,17,18]. Of note, silver stains may be falsely negative in cases with very few fungal organisms. Fourth, the absence of fungal organisms in the primary CT-guided core needle biopsy of the pulmonary lesion was most likely due to a sampling error. The finding of organizing pneumonia in the biopsy, which often surrounds necrotic lesions, supports this assumption.

Overall, this case stresses the importance of obtaining a careful travel history in a country where histoplasmosis is extremely rare and radiological findings can be mistaken for more common disorders, like malignant or autoimmune diseases. It also highlights the importance of knowledge about possible imported infectious diseases for effective diagnostics. With this case report, we also add further images to the limited database on [^18^F]FDG-PET/CT findings in patients with histoplasmosis and illustrate the potential diagnostic value of the “flip-flop fungus sign”.

## Figures and Tables

**Figure 1 diagnostics-11-00328-f001:**
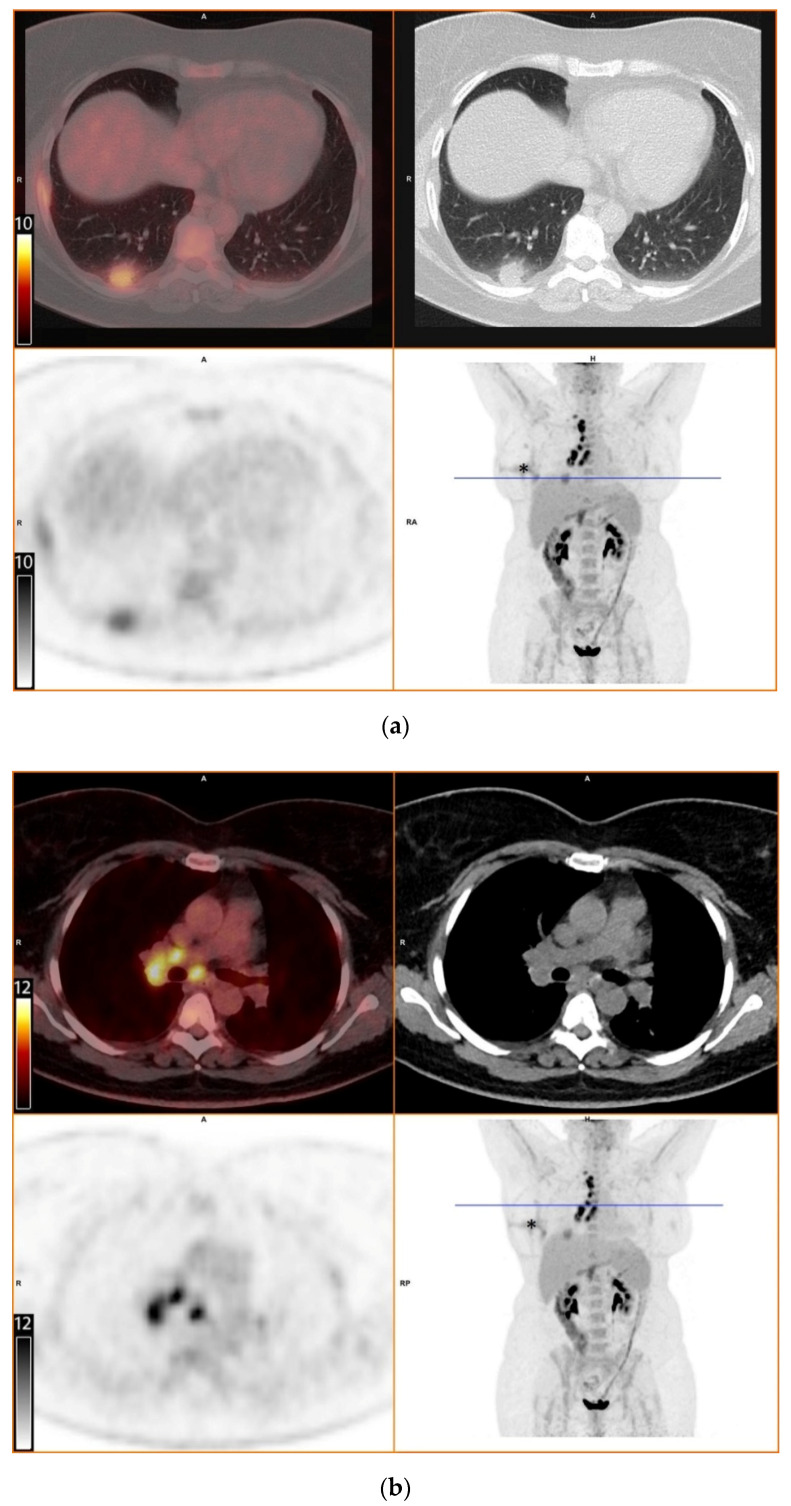
[^18^F]FDG-PET/CT. Upper left: PET and CT fusion images. Upper right: CT images. Lower left: PET images. Lower right: Maximum intensity projection. Blue line: Axial imaging level. Scale bars: Standardized uptake value. Star: Postinterventional diffuse uptake of the right thoracic wall after chest tube placement. (**a**) Moderate uptake in a solitary solid pulmonary lesion with irregular margins in the right lower lobe of the lung (**b**) Greater uptake in the draining lymph nodes right interlobar, right hilar and mediastinal (**c**) A further draining right supraclavicular lymph node showing high uptake as well (**d**) Moderate uptake in a precaval abdominal lymph node.

**Figure 2 diagnostics-11-00328-f002:**
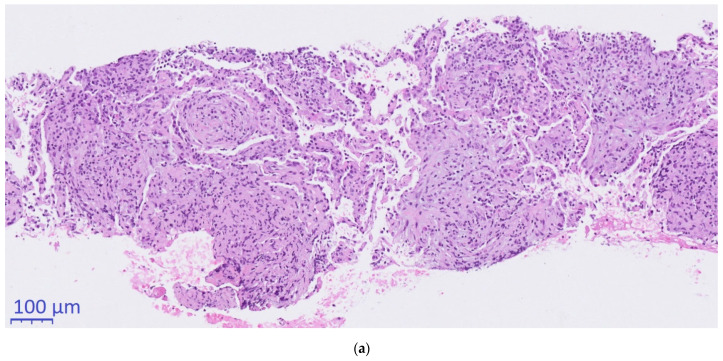
Histopathological evaluation. (**a**): 10×. (**b**): 2×. (**c**): 40×., (**a**,**b**): Hematoxylin & Eosin. (**c**): Grocott’s methenamine silver stain. (**a**) CT-guided core needle biopsy of the pulmonary mass showing organizing pneumonia (**b**) Supraclavicular lymph-node presenting with necrotizing granulomas (**c**) Typical ovoid *Histoplasma capsulatum* yeasts stained in black and thus made visible using Grocott’s methenamine silver staining.

**Table 1 diagnostics-11-00328-t001:** Positive flip-flop fungus sign criteria ^1^.

At least one pulmonary nodule is present	
	Solid or part solid (not ground glass)
	8–30 mm mean diameter
	Not necrotic, invasive, or calcified
	Any level of FDG activity data
At least one FDG-avid draining lymph node is present	
	SUVmax node > mediastinal blood pool
	Station 11 (ipsilateral), 4 (ipsilateral) or 7
At least one draining lymph node has ≥ SUVmax than the pulmonary nodule(s)	
Absence of FDG avid lesions worrisome for cancer	
	No obvious extrathoracic malignancy
	FDG avid lesions in reticuloenthelial system (lymph nodes, spleen, liver) permitted

^1^ Criteria by Nagelschneider and colleagues, Am J Nucl Med Mol Imaging 2017.

## Data Availability

Extracted patient data and images, as well as written informed consent from the patient are available from the corresponding author on request.

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
