# Peer review of "Pulmonary Histoplasmosis Mimicking Metastatic Lung Cancer: A Case Report"

_diagnostics, 2021, doi:10.3390/diagnostics11020328_

Round 1

Reviewer 1 Report

Histoplasma capsulatum is endemic to many parts of the world, in particular Latin America and the areas of the Midwestern and South central United States. Clinical manifestations of histoplasmosis are well described; the diagnosis of histoplasmosis cannot be achieved on the basis of clinical information alone, since there is significant overlap of histoplasmosis with other diseases. The definitive diagnosis requires the isolation of the H. capsulatum on specific culture media or the visualization of the yeast form in direct examination of clinical specimens using specific fungal staining techniques.

This case illustrates the difficulty in differentiating active histoplasmosis infection from malignancy of an immunocompetent patient living in a non-endemic region.

Histoplasmosis infection may radiologically resemble pulmonary malignancy, often causing a diagnostic dilemma. PET imaging is currently used for and considered accurate in the evaluation of pulmonary nodules. Autors show that FDG PET/CT could be useful for differentiating malignant lung and mediastinal lesions from benign granulomatous intrathoracic lesions, such as those associated with histoplasmosis.

Author Response

Dear Reviewer

We kindly thank you for your time and effort to review our manuscript.

We will upload a revised version of our manuscript with implementations of other reviewers suggestions.

Sincerely 
The authors

Reviewer 2 Report

It is an interesting well structured manuscript which reports the diagnostic utility of the flip flop fungus sign at the management of lung nodules. My only recommendation is  to add at the manuscript the flip flop fungus sign criteria first mentioned by Nagelschneider et al.

Author Response

Dear Reviewer

We kindly thank you for your time and effort to review our manuscript.

We will upload a revised version of our manuscript with implementations of your and other reviewers suggestions.

Sincerely 
The authors

Reviewer 3 Report

I thank the authors for presenting this very interesting case of pulmonary histoplasmosis with an unfortunately long time to diagnosis. It is of particular importance to everyone reading PET scans as it highlights particularly two very important things: First, it shows that the assessment of an FDG PET/CT is ultimately based on a probability calculation and that the exclusion/not considering of less likely differential diagnoses can have real clinical consequences. Second, this case highlights the importance of a complete anamnesis including travel history. 

The manuscript is well written and researched. There are only minor changes I would like the authors to implement before considering the work for publication:

  • In the affiliations line, please number the names in correct order, e.g. not X1,5 but X1,2
  • Change 2-[18F]FDG-PET-CT for [18F]FDG-PET/CT throughout the manuscript
  • Line 40: /CT
  • Line 41: change „greater FDG activity“ to „higher FDG uptake“
  • Line 55: change „, eg“ to „like“; Line 79 remove „eg“
  • Line 82: change „hyperintense“ with „hypermetabolic“
  • Line 167: change „FDG activity“ to „FDG uptake“
  • Line 177: change „moderately active precaval lymph node“ to „precaval lymph node with moderately elevated FDG uptake“
  • Add color bars / uptake bars to the PET and Fusion images

Author Response

(The authors gave the same response as above.)
